

# GCL-ALG: graph contrastive learning with adaptive learnable view generators

Yafang Li, Jie Kang, Zhihua Chu and Baokai Zu

College of Computer Science, Beijing University of Technology, Beijing, China

## ABSTRACT

Data augmentation is a pivotal part of graph contrastive learning, which can mine implicit graph data information to improve the quality of representation learning. Research on graph data augmentation has achieved promising results in recent years. However, existing graph contrastive learning methods are trapped in inherent predefined augmentation schemes, which greatly limits the generalization of augmentation methods. To this end, we propose a new adaptive original topology learnable data augmentation algorithm, Graph Contrastive Learning with Adaptive Learnable View Generators (GCL-ALG), to optimize the augmentation process and feature learning in an end-to-end self-supervised learning approach. Specifically, GCL-ALG introduces graph neural networks (GNN), graph attention modules and edge probability distributions to build a dual-level feature extraction framework to generate highly reliable representations, while integrating network science theory to selectively modify the strength of augmentation probabilities from node-level and edge-level, and then train dynamically learnable augmentation instances. Moreover, GCL-ALG designs multiple loss functions to drive the representation optimization to ensure that the generated graph representations are highly discriminative across different tasks. Extensive experiments are conducted on unsupervised learning, semi-supervised learning and transfer learning application tasks. The experimental results demonstrate the superior performance of the proposed GCL-ALG method on 16 benchmark datasets.

## INTRODUCTION

Graph representation learning has long been an important and challenging task (*Chen et al., 2020a*; *Hamilton, Ying & Leskovec, 2017*). In recent years, graph neural networks (GNNs) (*Wu et al., 2022*), which have shown better performance in graph representation learning, have garnered extensive attention in a wide range of application scenarios such as social network analysis (*Perozzi, Al-Rfou & Skiena, 2014*), recommendation system development (*He et al., 2023*; *Jiang, Huang & Huang, 2023*), knowledge graph analysis (*Huang et al., 2023*) and prediction of biochemical molecules (*Li, Huang & Zitnik, 2022*; *Senior et al., 2020*). However, most existing GNN models are trained in an end-to-end supervised manner (*Zhang et al., 2018*; *Kipf & Welling, 2016a*), which is both time-consuming and expensive to annotate. To address these issues, unsupervised contrastive learning designed GNN as self-supervised encoders has received attention from

Corresponding author
Baokai Zu, bzu@bjut.edu.cn

graph researchers (*Kipf & Welling, 2016b*; *Liu et al., 2022*, *2021*; *Zeng & Xie, 2021*). The unprecedented success of contrastive learning in computer vision (*Chen et al., 2020b*; *He et al., 2020*) and natural language processing (*Wang et al., 2021*; *Gao, Yao & Chen, 2021*; *Lu et al., 2022*) has inspired them to produce highly robust and transferable models by incorporating the idea of self-supervised learning.

In the process of constructing a contrastive model, data augmentation is considered as a critical factor, and the quality of the augmented views can even directly affect the final results of the experiment. Traditional data augmentation methods directly disrupt the original graph view by operations like node deletion, attribute masking, and edge perturbation to obtain augmented instances, whose limitations are: (1) The augmentation strategies are simple and fixed, and the augmentation process is blind, which greatly limits the diversity and expression ability of augmented instances, and may cause semantic drift at the same time. (2) The generation of augmented instances depends on predefined parameters or prior knowledge, which is difficult to automatically optimize for specific tasks. For example, GRACE (*Zhu et al., 2020*) combines random attribute masking and edge deletion to generate augmented samples, but the augmentation approach is coarse and relies on fixed augmentation parameters. GCA (*Zhu et al., 2021*) proposes an adaptive augmentation method to selectively remove structures or mask attributes in order to further ensure the basic semantics of the generated graphs. Nonetheless, relying solely on centrality characteristics to guide augmentation introduces a strong dependence on prior knowledge. GraphCL (*You et al., 2020*) tries to integrate multiple augmentation strategies to explore the best augmentation method adapted to the input graph, but static augmentation strategies would limit its expression space.

In recent years, automated graph augmentation has received considerable attention for achieving self-learning and optimizing augmentation operations to successfully address the limitations of manual design. JOAO (*You et al., 2021*) first migrates automated data augmentation to the graph domain, follows Bayesian theory to extract augmentation methods from a predefined augmentation pool, and optimizes the best solution automatically and iteratively *via* a max–min framework. GPA (*Zhang et al., 2024*) parameterizes the selection probabilities of augmented view pairs based on graph neural networks and adjusts the augmentation strategies through feedback from the bi-level optimization framework. But in both JOAO (*You et al., 2021*) and GPA (*Zhang et al., 2024*), the augmentation itself is not learnable. AD-GCL (*Suresh et al., 2021*) proposes an edge-level learnable data augmentation method based on Bernoulli distribution, but there is a single type of augmentation and the augmentation strategy is not substantially trained during the optimization of the algorithm. AutoGCL (*Yin et al., 2022*) automatically decides whether nodes or attributes should be retained, masked, or deleted by learning probabilities, but there is no edge-level augmentation, and learning probabilistic latent noise does not adequately ensure semantic consistency. RGCL (*Li et al., 2022*), based on the invariance principle, uses a rationale generator to discover discriminative semantic structures for data augmentation, but its node-level augmentation and focus on local information limit its ability to handle global structures and complex graphs. We find that these approaches lack task-driven signals in the augmentation learning process, making

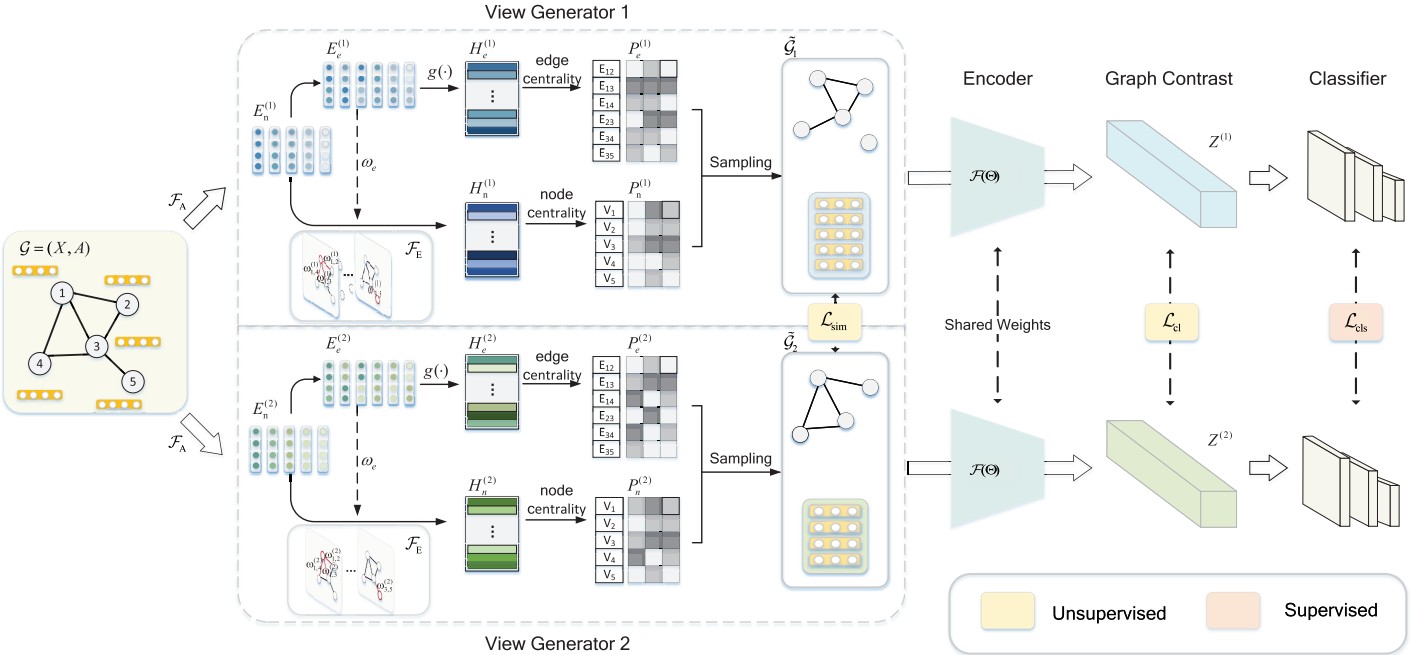

**Figure 1 Illustration of the GCL-ALG graph contrastive learning framework.** The GCL-ALG graph contrastive learning framework consists of three modules: view generators, graph feature encoder, and classifier. The GNN encoder $\mathcal{F}_A$ extracts deep node features from the original graph based on a self-attention mechanism, while $\mathcal{F}_E$ integrates edge load information to enhance node representations, where edge-level features are mapped to a unified representation space through the projection layer $g(\cdot)$. The view generators construct the sampling probabilities $P_{node}$ and $P_{edge}$ for nodes and edges by combining centrality metrics and sampling functions, with the sampling probabilities increasing from light to dark colors. The generated augmented views $\tilde{G}_1$ and $\tilde{G}_2$ are encoded by a shared encoder $\mathcal{F}(\cdot)$, and during training, contrastive learning is achieved by maximizing the consistency between their representations $Z^{(1)}$ and $Z^{(2)}$.

the augmentation goal unclear. Furthermore, given the non-Euclidean structural properties of graph data augmentation should be constructed on reliable representations, yet previous work neglects deep mining of graph representations prior to augmentation.

To solve the above issues, we propose a novel graph contrastive learning approach based on learnable data augmentation named Graph Contrastive Learning with Adaptive Learnable View Generators (GCL-ALG); the general framework is shown in Fig. 1. GCL-ALG first uses deep graph message passing to process the implicit structural relationships and attribute information of input graphs, obtaining highly secure and reliable graph representations. Specifically, a dual-level feature extraction framework is introduced into the view generators to complementarily extract graph information. The first level combines self-attention networks with GNN encoder to focus on the areas of information propagation, while the second level integrates edge load information to extract high-order global representations. Subsequently, centrality metrics are used as augmentation guidance signals, and the reconstruction capabilities of autoencoders are leveraged to construct learnable view generators that produce high-quality augmented instances. In particular, our proposed method comprehensively studies the augmentation patterns of nodes and edges, utilizing centrality to refine the feature distributions to minimize the impact of interference noise. Finally, the trained augmented instances are

projected into low-dimensional graph representations *via* graph encoders for downstream graph classification, and view generators and contrastive learning framework are jointly optimized by multiple objective functions to obtain a highly robust and learnable graph contrastive learning algorithm.

In summary, the contributions of this article are as follows:

- We propose a contrastive comparative learning framework based on highly adaptive and learnable view generators, which can automatically obtain augmented views according to the intrinsic properties of the graph.
- The proposed GCL-ALG framework utilizes attention mechanisms and edge weights to capture important feature information, and trains view generators from dual perspectives of nodes and edges to learn a comprehensive representation of the graph structure.
- We extensively evaluate GCL-ALG on 15 different benchmark datasets to validate the performance on unsupervised learning, transfer learning and semi-supervised learning graph classification tasks. Experimental results demonstrate that GCL-ALG exhibits outstanding performance in terms of average relative improvement compared to state-of-the-art baseline methods. In addition, we conduct ablation experiments to analyze the contribution of the dual-level feature extraction framework in the performance of the algorithm.

## RELATED WORK

### Contrastive learning

Contrastive learning (CL) has made promising achievements in image representation learning (*Chen et al., 2020b*; *Zhu et al., 2021*) thanks to its powerful self-supervised learning ability. CL learns robust representations by pulling in positive pairs and pushing out negative pairs, and utilizing the relative relationship between samples to determine semantic associations. Therefore, many researchers attempted to apply contrastive methods to graph domain in order to extract meaningful graph structure semantic relationships. However, the intrinsic structural complexity of graph data also poses greater challenges. Previous works have been improved from several perspectives such as different contrast angles, augmentation strategies, positive and negative sample selection, *etc*. DGI (*Velickovic et al., 2018*) is driven by the mutual information estimation proposed by MINE (*Belghazi et al., 2018*) to maximize the mutual information between the global graph embedding and the local node embedding. GMI (*Peng et al., 2020*) learns node representation vectors by maximizing the mutual information between the encoder input representation and the output hidden representation. MVGRL (*Hassani & Khasahmadi, 2020*) constructs node-level and graph-level augmented instances based on graph diffusion and first-order neighborhoods, and proposes a cross-view multi-scale contrastive learning approach. GRACE (*Zhu et al., 2020*) generates two augmented graphs by removing edges and masking attributes, focusing on maximizing the consistency of node embedding between broken graphs. Graph contrastive representation learning with adaptive

augmentation (GCA) (*Zhu et al., 2021*) builds upon GRACE (*Zhu et al., 2020*) and proposes adaptive augmentation based on the topology and attributes of the graph, enabling it to dynamically adjust the augmentation strategy according to the specific features of the input graph. GraphCL (*You et al., 2020*) designs a data augmentation method with a combination of four approaches: node removal, edge perturbation, subgraph sampling and feature masking, revealing that the selection of augmentation strategies needs to be adapted to the specific traits of different tasks. GCP (*Adjeisah et al., 2024*) also adopts diverse augmentation strategies to generate multi-view graph representations, and leverages pre-training to automatically identify the optimal augmentation configuration. However, *Xia et al. (2022)* question the necessity of data augmentation by proposing SimGRACE, which perturbs the encoder to generate contrastive views directly from the original graph, thereby avoiding semantic bias introduced by manual augmentations while achieving competitive performance across multiple benchmark tasks. NCAGC (*Wang et al., 2023*) uses k-nearest neighbors (KNN) to construct intra-cluster and inter-cluster neighborhood contrastive relations to guide the learned feature representations towards clustering. SUGRL (*Mo et al., 2022*) exploits the complementary information between structural and neighbor information to generate positive samples, and designs a multi-objective contrastive loss to enforce a closer relation to positive samples and a further divergence from negative samples.

We note that the augmentation operations in the above approaches obtain augmentation instances based on simple predefined parameters violently destroying the original graph, which may cause severe semantic loss leading to unstable quality of the generated views. To tackle these concerns, GCL-ALG builds learnable view generators to automatically acquire valid augmentation instances and provide rich augmentation variance.

## Automated data augmentation

Data augmentation enriches training samples while maintaining label consistency. As demonstrated by *Adjeisah et al. (2023)*, augmentation strategies play a pivotal role in enhancing the robustness and generalization capabilities of graph neural networks, especially in data-scarce scenarios. However, most existing methods still rely on manually predefined augmentation operations, which are limited in their adaptability to diverse task requirements. In contrast, automated data augmentation has emerged as a promising alternative, enabling dynamic strategy optimization based on both input data characteristics and task objectives, thus garnering increasing research attention. In the computer vision field, AutoAugment (*Cubuk et al., 2019*) first proposes to utilize reinforcement learning to achieve automated data augmentation. *Cubuk et al. (2020)* propose the RandomAugment (*Cubuk et al., 2020*) method using a parameter-free procedure instead of learning strategies and probabilities, successfully balancing data augmentation diversity and search costs. BO-Aug (*Zhang et al., 2023*) proposes to explore optimal augmentation schemes in the space of predefined data augmentation strategies using Bayesian optimization techniques.

These automated image augmentation methods inspire graph contrastive learning. JOAO (*You et al., 2021*) learns the sampling distribution of predefined augmentation operations on the top of GraphCL (*You et al., 2020*), but the augmentation itself is not learnable. AD-GCL (*Suresh et al., 2021*) is motivated by adversarial learning and proposes a learnable edge-level data augmentation. AutoGCL (*Yin et al., 2022*) proposes learnable view generators driven by learning probabilities. RGCL (*Li et al., 2022*) utilizes a learnable rationale generator to identify and preserve discriminative node sets in the graph, combines complement sampling to construct non-discriminative structures, and optimizes augmented views through adversarial training. HAGCL (*Chen, Ren & Yong, 2023*) constructs dedicated view generators from feature and edge perspectives to achieve mixed data augmentation. GCC-LDA (*Yang et al., 2022*) proposes to automatically generate structure-based and attribute-based augmentation with attention mechanisms. NCLA (*Shen et al., 2023*) proposes a learnable augmentation based on multi-head attention mechanisms. However, the augmentation perspectives in previous studies are still not comprehensive, and the augmentation process lacks guidance signals. In this work, we performed fine-grained augmentation separately on the graph constituents- nodes and edges, introduce network centrality as guidance signals so that augmentation instances are oriented to key elements, and optimize the learnable augmentation parameters through an autoencoder to dynamically balance semantic preservation and view variability.

# PROPOSED METHOD

## Notions and preliminaries

Given a graph $\mathcal{G} \in \{\mathcal{G}_m : m \in M\}$ from the dataset of $M$ graphs. Define graph $\mathcal{G} = (\mathcal{V}, \mathcal{E})$, considering $\mathcal{V} = \{v_1, v_2, \cdots, v_N\}$ and $\mathcal{E} \subseteq \mathcal{V} \times \mathcal{V}$ represent the node set and edge set respectively. $\boldsymbol{X} \in \mathbb{R}^{N \times F}$ and $\boldsymbol{A} \in \{0, 1\}^{N \times N}$ denote the feature matrix and the adjacency matrix, where $x_i \in \boldsymbol{X}$ indicates the feature vector of node $v_i$ and $A_{ij} = 1$ if $(v_i, v_j) \in \mathcal{E}$ otherwise $A_{ij} = 0$. $\mathcal{N}(v_i) = \{v_j \mid j \neq i, A_{ij} = 1\}$ indicates a set of 1-top neighbors of $v_i$. $\boldsymbol{D} = diag(d_1, d_2, \ldots, d_N) \in \mathbb{R}^{N \times N}$ denotes the degree matrix, where $d_i = \sum_{(v_i, v_j) \in E} A_{ij}$. Input graphs to two separate view generators generate views $\tilde{G}_1, \tilde{G}_2$, where the process base node features $\boldsymbol{h}_i$ and edge features $\boldsymbol{h}_{ij}$ count sampling to generate augmentation strategy matrices $\boldsymbol{S_1} \in \mathbb{R}^N$, $\boldsymbol{S_2} \in \mathbb{R}^N$. Through the graph encoder $\mathcal{F}(\cdot)$, both node-level representations $\boldsymbol{H}_n^1 \in \mathbb{R}^{N \times F'}$, $\boldsymbol{H}_n^2 \in \mathbb{R}^{N \times F'}$ and graph-level representations $\boldsymbol{H}_g^1 \in \mathbb{R}^{D \times F''}$, $\boldsymbol{H}_g^2 \in \mathbb{R}^{D \times F''}$ are produced. Detailed information is shown in Table 1.

- **Graph neural networks.** Real-world data structures are very complex and are gradually characterized by graph structure. *Scarselli et al. (2008)* proposed the GNN (*Scarselli et al., 2008*) as an effective method for the analysis of graph-structured data. In this work, we focus on GNN as the backbone network of the encoder. For a given graph $\mathcal{G}$, each node $v_i$ has a feature distribution $\boldsymbol{h}_i$, initialized as $\boldsymbol{h}_i^{(0)} = \boldsymbol{x}_i$, then $v_i$ at the $k$-th GNN message propagation can be stated as,

$$\boldsymbol{a}_i^{(k)} = AGGREATION^{(k)}(\{\boldsymbol{h}_i^{(k-1)} : i \in \mathcal{N}(i)\}), \tag{1}$$

**Table 1 Description of notations in the article.**

| Notion | Meaning |
|---|---|
| $V$ | Node set |
| $E$ | Edge set |
| $X \in \mathbb{R}^{N \times F}$ | Original feature matrix |
| $A \in \mathbb{R}^{N \times N}$ | Original adjacency matrix |
| $D \in \mathbb{R}^{N \times N}$ | Degree matrix |
| $\mathcal{N}(i)$ | 1-top neighbors set of node $v_i$ |
| $E_n^k \in \mathbb{R}^{N \times F^k}$ | Node embedding in $k$-th convolution |
| $E_n \in \mathbb{R}^{N \times F^k}$ | Embedding processed through the attention mechanism |
| $H_n \in \mathbb{R}^{T_n \times F'}$ | Final node embedding |
| $H_e \in \mathbb{R}^{T_e \times C}$ | Final edge embedding |
| $Z \in \mathbb{R}^{Q \times D}$ | Graph embedding |
| $S \in \mathbb{R}^N$ | Sample matrix |

$$h_i^{(k)} = COMBINE^{(k)}(h_i^{(k-1)}, a_i^{(k)}). \tag{2}$$

where both $AGGREGATE(\cdot)$ and $COMBINE(\cdot)$ are trainable functions, serving the purpose of aggregating and updating messages respectively. After a specified number of iterations, the READOUT() function is used to aggregate embeddings from all nodes, generating a graph-level representation. Then, this representation is converted into an output suitable for downstream tasks through a multi-layer perceptron (MLP), denoted as $z_G$:

$$z_G = MLP\left(READOUT(\{h_i^{(k)}, v_i \in V\})\right). \tag{3}$$

- **Gumbel-Softmax.** We employ Gumbel-Softmax (*Jang, Gu & Poole, 2016*) as our sampling function in this work, which can convert a discrete random variable into a differentiable continuous distribution *via* reparameterization, and enabling backpropagation in neural networks. Assume the network propagation process yields a collection of discrete feature distributions $\pi_1, \pi_2, \ldots, \pi_n$, where $\pi_i$ denotes the $i$-th edge or node feature, $\sum_i^n \pi = 1$, then the sampling result $z_i$ can be represented as,

$$z_i = \frac{exp\left(g_i + \log \pi_i\right)/\tau}{\sum_j^n exp\left(g_j + \log \pi_j\right)/\tau}, \tag{4}$$

where $\tau$ represents temperature coefficient, $g_i$ is a pre-collected sample from the Gumbel (0,1) distribution. It can be observed that the higher the probability of a particular feature distribution, the more likely it is to be selected during Gumbel-Softmax sampling. For this purpose, we generate prior probabilities from the original graph to adjust the sampling distribution in order to maximize the retention of important information in the graph.

## Learnable view generator

In this section, we provide a detailed description of the view generators proposed in GCL-ALG, which includes how to explore graph features, how to incorporate adaptive metrics, and how to implement feature sampling.

- **Dual-layer feature extraction.** Graph neural networks often struggle to simultaneously capture local connectivity patterns and global topological structures in complex graph tasks. To address this issue, we propose a dual-layer feature extraction framework that hierarchically models structural information at different granularities. By leveraging a local-global complementary feature extraction scheme, the framework enhances the representational coverage and structural sensitivity, thereby improving the model's adaptability to diverse types of graph data.

The view generators integrate a self-attention-based encoder $\mathcal{F}_A$ and an edge-based encoder $\mathcal{F}_E$, with the primary objective of collaboratively extracting critical structural information from the graph.

**Self-Attention-based GNN encoder.** The $\mathcal{F}_A$ encoder integrates the strong structural modeling capacity of graph isomorphism network (GIN) with the selective feature extraction capability of the attention mechanism, enabling it to effectively capture salient node features within local neighborhoods. Assuming a total of $K$ GIN layers are used, a new node feature matrix $E_n^k \in \mathbb{R}^{N \times F^k}$ is generated after each convolution layer. We denote [;] is the concatenation operator and the $k$-th iteration of the iterative aggregation update formula for node $v_i$ as:

$$h_i^{(k)} = COMBINE^{(k)}\left(h_i^{(k-1)}, a_i^{(k)}\right), \tag{5}$$

$$a_i^{(k)} = AGGREATION^{(k)}\left(\{h_i^{(k-1)} : i \in \mathcal{N}(i)\}\right), \tag{6}$$

$$E_n^k = \left[h_1^{(k)}; h_2^{(k)}; \dots ; h_N^{(k)}\right]^T. \tag{7}$$

Inspired by Transformer (*Vaswani et al., 2017*), we seamlessly integrate a self-attention network module after each GIN layer, consciously guiding the network propagation process to keenly capture more crucial node-level information, thereby elevating the model's learning capability to new heights. The hidden layer features $E_n^k$ are mapped to three different vector spaces: $Q \in \mathbb{R}^N \times \mathbb{F}^k$, $K \in \mathbb{R}^{N \times F^k}$, $V \in \mathbb{R}^{N \times F^k}$, which respectively represent the query matrix, key matrix and value matrix.

$$Q = W_q(X^k)^T, \tag{8}$$

$$K = W_k(X^k)^T, \tag{9}$$

$$V = W_v(X^k)^T, \tag{10}$$

where $W_q \in \mathbb{R}^{F^k \times F^k}$, $W_k \in \mathbb{R}^{F^k \times F^k}$, $W_v \in \mathbb{R}^{F^k \times F^k}$ are the initialized weight matrices, and the node embedding $E_n \in \mathbb{R}^{N \times F^k}$ after the attention mechanism can be expressed as:

$$E_n = softmax\left(\frac{QK^T}{\sqrt{F^k}}\right)V. \tag{11}$$

**Edge-based GNN encoder.** To compensate for the $\mathcal{F}_A$ encoder's limited understanding of the overall graph structure, the $\mathcal{F}_E$ encoder is introduced to incorporate higher-order structural information. $\mathcal{F}_E$ leverages edge weight to measure the importance of edges in global information flow, effectively capturing the macro structure and global dependencies

of the graph. We first concatenate the node features to generate preliminary edge features $E_e$, and then employ an MLP to parameterize the edge features and extract weight information,

$$w_{ij} = MLP\left(\left[\boldsymbol{h}_i^{(K)}, \boldsymbol{h}_j^{(K)}\right]\right), \tag{12}$$

where $\boldsymbol{h}_i^{(K)}, \boldsymbol{h}_i^{(K)} \in \boldsymbol{H_n}$ denote the feature vectors of nodes $v_i$ and $v_j$ respectively after $\mathcal{F}_A$ encoder.

To learn the edge weights, $w_e$ is normalized to a continuous variable between $[0, 1]$. Specifically, we employ $p_{ij} = sigmoid((log\delta - log(1-\delta) + w_e)/\tau)$ to represent the weight of the edge that links nodes $v_i$ and $v_j$, where $\delta \sim Uniform(0, 1)$ and $\tau$ is a temperature hyperparameter. Eventually, $\mathcal{F}_E$ adjusts the aggregation process of node features using this weight information, giving greater emphasis to edges with higher weights, thus improving the model's emphasis on the edges. In formal terms, the formula for aggregated edge weights can be denoted as:

$$\boldsymbol{h}_i^{(k)} = COMBINE^{(k)}\left(\boldsymbol{h}_i^{(k-1)}, \boldsymbol{a}_i^{(k)}\right), \tag{13}$$

$$\boldsymbol{a}_i^{(k)} = AGGREATION^{(k)}\left(\left\{p\boldsymbol{h}_i^{(k-1)} : i \in \mathcal{N}(i)\right\}\right). \tag{14}$$

Then we obtain the final node feature matrix $\boldsymbol{H_n} = \left[\boldsymbol{h}_1^{(K)}; \boldsymbol{h}_2^{(K)}; \ldots ; \boldsymbol{h}_N^{(K)}\right]^T \in \mathbb{R}^{D \times F'}$, $F' \ll F$. The raw edge feature $E_e$ passes through the projection layer to capture high-order crossed features, obtaining the final edge feature. He represented by the formula $\boldsymbol{H_e} = g(\boldsymbol{E_e})$, where $g(\cdot)$ is the projection function.

- **Node sampling.** With the above procedures, we have successfully generated graph embeddings that encompass adaptive and crucial features. Now, we need to further adaptively sample node features and edge features. To be more specific, we again cut from the perspective of the original graph, generate prior probabilities for nodes and edges based on the original graph, and incorporate these probabilities into the subsequent sampling process. The underlying logic of this strategy is to guide the sampling function to selectively favor nodes and edges in the original graph that are considered essential, thereby avoiding a completely random or uniform sampling of all elements.

  We leverage the Gumbel-Softmax (*Jang, Gu & Poole, 2016*) reparametrization trick to generate a differentiable probability distribution, thus enabling the optimization of the sampling operation through gradient descent. Additionally, the introduction of Gumbel noise maintains a certain level of randomness, which is beneficial in preventing the model from over-concentrating on specific features, thereby mitigating the risk of under-generalization. Figure 2 takes node sampling as an example to describe in detail the feature correction and sampling process. The sampling probability of node $v_i$ can be expressed as:

$$P_i = Gumbel\_Softmax(Node\_Cent(v_i)\boldsymbol{h_i}), \tag{15}$$

  where $Node\_Cent(\cdot)$ is the node centrality measure formula, $\boldsymbol{h_i} \in \boldsymbol{H_n}$ is the feature vector of node $v_i$, and $Node\_Cent(\cdot)\boldsymbol{h_i}$ denotes the adaptive sampling probability. With a

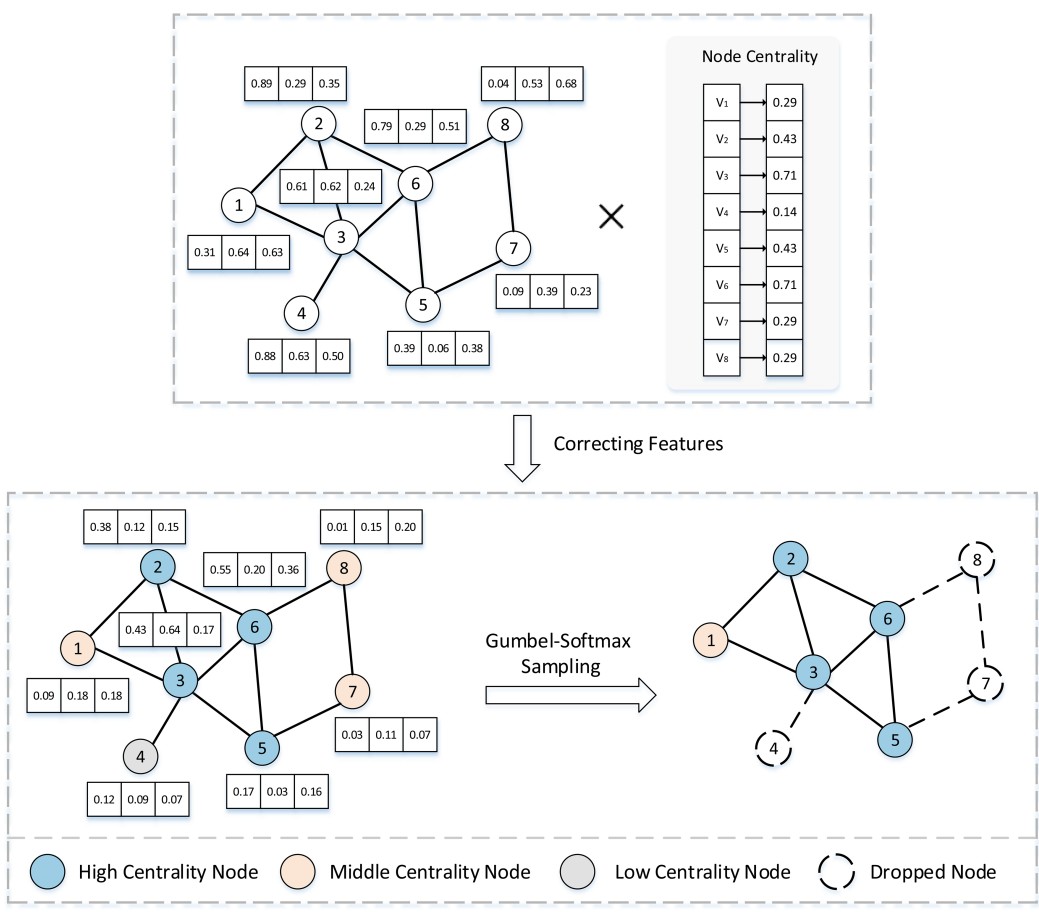

**Figure 2 Diagram of node feature correction and sampling.**

combination of efficiency and performance considerations, we adopt node degree centrality as the metric function, which is formulated as:

$$Node\_Cent(v_i) = \frac{d_i}{N-1}. \tag{16}$$

In this context, $N$ denotes the total number of nodes in the network which node $v_i$ belongs. Degree centrality is the most straightforward and broadly applied measure of node centrality in network analysis. This metric posits that nodes with more associated nodes are considered more important. For instance, in social networks, nodes represent users and edges represent connections between users. Users with a higher degree centrality tend to have greater influence.

- **Edge sampling.** The edge sampling process is akin to node sampling, which aims to filter out edges with infrequent interactions between nodes. We utilize MLP to reduce the dimensionality of the generated edge features to prevent too much information from being lost in subsequent sampling, while obtaining a representative edge feature matrix

$H_e$. Then, by performing a Hadamard transformation between $H_e$ and the learned edge importance distribution from the original graph, we obtain an adaptive distribution matrix for the edges. Finally, we use the Gumbel_Softmax function to sample from the adaptive probability distribution and select the edges to be retained. In this process, $e_{ij}$ represents the edge connecting nodes $v_i$ and $v_j$, and $h_{ij} \in H_e$ represents the feature vector of edge $e_{ij}$, then we have:

$$P_{ij} = Gumbel\_Softmax(Edge\_Cent(e_{ij})h_{ij}), \tag{17}$$

where $Edge\_Cent(\cdot)$ indicates edge centrality formula. We choose the average degree centrality of the two nodes linked by the edge as the edge centrality metric, which can better preserve the stability of the network structure.

$$Edge\_Cent(e_{ij}) = \frac{Node\_Cent(v_i) + Node\_Cent(v_j)}{2}. \tag{18}$$

Through the aforementioned adaptive encoding and sampling strategies, we successfully preserve the key elements of the graph's inherent attributes, which form the foundation for deep integration of the graph.

## Architecture of contrastive learning

In this article, the proposed graph contrastive learning framework adheres to the well-known InfoMin (*Tian et al., 2020*) principle, aiming to maximize the mutual information between the augmented views and the input graph while minimizing the mutual information between generated views. Following the settings of AutoGCL (*Yin et al., 2022*), we adopt three loss functions to constrain the contrastive learning framework: contrastive loss, similarity loss and classification loss.

Drawing from the previous works (*Chen et al., 2020b*; *You et al., 2020*), we use the normalized temperature-scaled cross entropy loss (NT-Xent) (*Sohn, 2016*) as our contrastive loss function to force the positive sample pairs to draw closer while separating the negative samples. Randomly select a batch comprising $N$ graphs and obtain $2N$ views *via* two independent view generators. Arbitrarily specify $z_{1,i} \in Z^{(1)}, z_{2,j} \in Z^{(2)}$ as the generated views from different view generators respectively, and treat the two as a pair of positive samples. Negative samples are naturally defined as views from the other $2(N-1)$ views. Formally, we define the cosine similarity function $\theta(z_{1,i}, z_{2,j}) = \frac{z_{1,i} \cdot z_{2,j}}{\|z_{1,i}\|_2 \cdot \|z_{2,j}\|_2}$. Set the pairwise objective for each positive pair $(z_{1,i}, z_{2,j})$ as $\ell(i,j)$, then the overall objective function can be expressed as $\mathcal{L}_{cl}$, where $\tau$ denotes the temperature parameter.

$$\ell(i,j) = -\log \frac{e^{\theta(z_{1,i},z_{2,j})/\tau}}{\sum_{\substack{m,n=1 \\ m \leq n}}^{N} e^{\theta(z_{1,i},z_{1,m})/\tau} + e^{\theta(z_{1,i},z_{2,n})/\tau}}, \tag{19}$$

$$\mathcal{L}_{cl} = \frac{1}{2N} \sum_{k=1}^{N} [\ell(2k-1, 2k) + \ell(2k, 2k-1)] \tag{20}$$

In each iteration of GCL-ALG, the original graph $G$ is first input into the view generators, which produce two adaptive augmentation strategy matrices (or sampling matrices) $S_1$ and $S_2$, along with their corresponding augmented views $\tilde{G}_1$ and $\tilde{G}2$. To avoid the issue of view collapse, we aim to ensure sufficient diversity between the two augmented views. To this end, we introduce a view similarity loss function $L_{sim}$ in the semi-supervised setting, defined as:

$$\mathcal{L}_{sim} = 1 - \|S_1 - S_2\|_2^2. \tag{21}$$

The loss term penalizes the similarity between the augmentation strategies, explicitly encouraging the generation of diverse views. Since this mechanism significantly intervenes in the view generation process, it may disrupt the semantic consistency of the original graph structure in unsupervised tasks. Therefore, this loss is only used in the semi-supervised setting.

Finally, to further enhance the model's ability to fit the known labels and improve the discriminative power of the learned representations, we introduce a graph classification loss function $L_{cls}$ under the semi-supervised setting. Specifically, the original graph $G$ and its two augmented views $\tilde{G}_1$ and $\tilde{G}2$ are jointly fed into the classifier $C$, and the standard cross-entropy function $\ell_{cls}$ is applied to compute the prediction loss with respect to the corresponding label $y$. The loss is formally defined as follows:

$$\mathcal{L}_{cls} = \frac{1}{3}[\ell_{cls}(C(\tilde{G}_1), y) + \ell_{cls}(C(\tilde{G}_2), y) + \ell_{cls}(C(\tilde{G}), y)]. \tag{22}$$

## The time complexity

The GCL-ALG algorithm consists of three main parts. The first part is view generator based on dual-level feature extraction, the first level of feature extraction is based on the self-concerned encoder corresponding time complexity is $\mathcal{O}(l_a(|E|F + N^2F))$, where $l_a$ denotes the number of hidden layers. The second level of feature extraction fuses the edge weight information mined by the MLP into the GIN message propagation process with time complexity of $\mathcal{O}(l_e NH^2)$, where $H$ is the hidden layer dimension, and $l_e$ is the number of hidden layers. The second part is a graph encoder with a GIN backbone whose time complexity is compressed to $\mathcal{O}(l_g(|E|H + NH))$, where $l_g$ denotes the number of hidden layers of the graph encoder. The third part is the downstream classifier, the unsupervised representation learning uses SVM as the classifier with $\mathcal{O}(ND^2)$ time complexity, the semi-supervised task predicts the category to which the graph belongs based on the ResGCN network with $\mathcal{O}(l_s|E|D)$ time complexity, and the migration learning generates the classification results with the GIN network with $\mathcal{O}(l_t ED^2)$, where $l_s$, $l_t$ represent the number of hidden layers for semi-supervised and migration learning classifiers, respectively. Therefore, the total time complexity of unsupervised learning, semi-supervised learning and transfer learning are compressed as: $\mathcal{O}(|E|(F + H) + N(F^2 + H^2 + D^2))$, $\mathcal{O}(|E|(F + H) + N(F^2 + H^2) + l_s|E|D)$, $\mathcal{O}(|E|(F + H) + N(F^2 + H^2) + l_t|E|D)$, respectively. The detailed pseudo-code description of GCL-ALG is given in Algorithm 1.

---

> **Algorithm 1** Training learnable view generators with GCL-ALG principle.
>
> **Require:** Graph Feature Encoder $f_\theta$; View Generators $Gen_\Phi$, $Gen_\Psi$; Iteration Number $I$; Graph $\mathcal{G} \in \{\mathcal{G}_m : m \in M\}$; Initial Feature Matrix $\boldsymbol{X}$, $\boldsymbol{x_n} \in \boldsymbol{X}$; Learning Rate $\beta$; Hyperparameters $\lambda$, $\mu$.
>
> **Ensure:** Trained Graph Feature Encoder $f_\theta$ and View Generators $Gen_\Phi$, $Gen_\Psi$.
>
> 1:     **while** $i = 1$ *to* $I$ **do**
> 2:        **for** *sampled minibatch* $\{\mathcal{G} = (V, E), \boldsymbol{x_n} : n = 1, 2, \ldots, N\}$ **do**
> 3:        **for** $n = 1$ *to* $N$ **do**
> 4:           set centrality matrix $\boldsymbol{C_{node}}$, $\boldsymbol{C_{edge}}$ of $\mathcal{G}$ using Eqs. (16) and (18);
> 5:           $\boldsymbol{view}_{1,n} = Gen_\Phi(\boldsymbol{x_n}, \boldsymbol{C_{node}}, \boldsymbol{C_{edge}})$;
> 6:           $\boldsymbol{view}_{2,n} = Gen_\Psi(\boldsymbol{x_n}, \boldsymbol{C_{node}}, \boldsymbol{C_{edge}})$;
> 7:           $\boldsymbol{z_{1,n}} = f_\theta(\boldsymbol{view}_{1,n})$;
> 8:           $\boldsymbol{z_{2,n}} = f_\theta(\boldsymbol{view}_{2,n})$;
> 9:        **end for**
> 10:        **if** unsupervised task or transfer task
> 11:           compute the contrastive objective for minibatch with Eq. (19);
> 12:           $\mathcal{L} = \mathcal{L}_{cl}(\boldsymbol{view}_{1,n}, \boldsymbol{view}_{2,n})$;
> 13:        **end if**
> 14:        **if** semi-supervised task **then**
> 15:           compute the contrastive loss, similarity,and classification loss from Eqs. (19), (21) and (22);
> 16:           $\mathcal{L} = \mathcal{L}_{cl}(\boldsymbol{view}_{1,n}, \boldsymbol{view}_{2,n}) + \mathcal{L}_{sim}(\boldsymbol{z_{1,n}}, \boldsymbol{z_{2,n}}) + \mathcal{L}_{cls}(f_\theta(\boldsymbol{x_n}), \boldsymbol{z_{1,n}}, \boldsymbol{z_{2,n}})$;
> 17:        **end if**
> 18:        Update the params of $f_\theta$, $Gen_\Phi$ and $Gen_\Psi$ *via* gradient descent;
> 19:        $\theta \Leftarrow \theta - \beta\nabla_\theta(\mathcal{L})$, $\Phi \Leftarrow \Phi - \beta\nabla_\Phi(\mathcal{L})$, $\Psi \Leftarrow \Psi - \beta\nabla_\Psi(\mathcal{L})$;
> 20:        **end for**
> 21:        Return $f_\theta$, $Gen_\Phi$, $Gen_\Psi$;
> 22:    **end while**

## EXPERIMENT

### Datasets description

Our experiments include generalized datasets from multiple domains. For supervised and semi-supervised learning, we use the TUDataset (*Morris et al., 2020*), which provides a wide range of graph datasets, including but not limited to protein datasets and social network datasets. In addition, we predict chemical molecule properties and biological protein functions through transfer learning, in which we adopt the ChEMBL dataset (*Mayr et al., 2018*; *Gaulton et al., 2011*) containing 456K complex biochemical molecules as the pre-training dataset, fine-tuning the small molecule dataset of the MoleculeNet benchmark (*Wu et al., 2018*). We perform statistics based on dataset categories as shown in Table 2. The TuDataset download link is https://chrsmrrs.github.io/datasets/docs/datasets, the

**Table 2 Summary of the TUDataset benchmark dataset (*Morris et al., 2020*) for unsupervised and semi-supervised learning experiments and datasets of biochemical molecules from *Mayr et al. (2018)*, *Gaulton et al. (2011)*, and *Hu et al. (2019)* for transfer learning.**

| Dataset | Utilization | #Graphs | Avg.#Nodes | Avg.#Edges | #Classes |
|---|---|---|---|---|---|
| Social networks | | | | | |
| IMDB-B | unsupervised | 1,000 | 19.8 | 96.53 | 2 |
| COLLAB | unsupervised/semi-supervised | 5,000 | 74.5 | 2,457.78 | 3 |
| REDDIT-M-5K | unsupervised/semi-supervised | 4,999 | 508.8 | 594.87 | 5 |
| Biochemical molecules | | | | | |
| MUTAG | unsupervised | 188 | 17.93 | 19.79 | 2 |
| PROTEINS | unsupervised/semi-supervised | 1,113 | 39.06 | 72.82 | 2 |
| NCI1 | unsupervised/semi-supervised | 4,110 | 29.87 | 32.30 | 2 |
| DD | unsupervised/semi-supervised | 1,178 | 284.32 | 715.66 | 2 |
| chembl_filtered | Pre-training | 430,702 | 22.75 | 27.86 | 2 |
| Chemical molecules | | | | | |
| BBBP | Finetuning | 2,039 | 24.06 | 25.95 | 2 |
| Tox21 | Finetuning | 7,831 | 18.57 | 19.29 | 3 |
| ToxCast | Finetuning | 8,576 | 18.78 | 19.26 | 3 |
| SIDER | Finetuning | 1,427 | 33.64 | 35.36 | 2 |
| ClinTox | Finetuning | 1,477 | 26.16 | 27.88 | 2 |
| MUV | Finetuning | 93,087 | 24.23 | 26.28 | 3 |
| HIV | Finetuning | 41,127 | 25.51 | 27.47 | 2 |
| BACE | Finetuning | 1,513 | 34.09 | 36.86 | 2 |

ChEMBL dataset can be found at https://www.ebi.ac.uk/chembl, and the MoleculeNet dataset can be downloaded from https://moleculenet.org/datasets-1. The code for our implementation can be found in https://github.com/lincyli/GCL-ALG.

## Baselines

The effectiveness of the proposed method is verified by comparing it with the following 13 baseline approaches on three tasks: unsupervised learning, semi-supervised learning, and transfer learning.

- InfoGraph (*Sun et al., 2019*): A method for learning graph representations based on maximizing mutual information.
- GraphCL (*You et al., 2020*): A method that explores diverse features through combinatorial graph augmentations.
- GCA (*Zhu et al., 2021*): The method leverages network centrality and adaptive data augmentation to retain important elements.
- JOAO (*You et al., 2021*): The method constructs an augmentation-aware projection head to find the optimal augmentation strategy from a predefined pool of augmentations.
- AD-GCL (*Suresh et al., 2021*): An augmentation method that employs adversarial learning to delete edges, based on a max-min mutual information framework.

- AutoGCL (*Yin et al., 2022*): A method that uses data-driven active learning to adjust node dropout and feature masking probabilities in order to maximize retention of the original semantics.
- SimGRACE (*Xia et al., 2022*): An Augmentation-Free graph contrastive learning framework *via* Gaussian-Perturbed Encoders.
- RGCL (*Li et al., 2022*): A rationale-aware graph contrastive learning framework grounded in the principle of invariance.
- GPA (*Zhang et al., 2024*): A graph contrastive learning method with personalized augmentation selector for graph-level adaptive augmentation.
- Infomax (*Velickovic et al., 2018*): An approach for learning node representations by maximizing local-global mutual information.
- EdgePred, AtrrMasking, and ContextPred: Node-level pre-training methods proposed by Pretrain-GNN (*Hu et al., 2019*), involving original edge prediction, learning node/ edge attributes, and predicting surrounding graph structures with subgraphs for training GNN to explore deep representations.

## Experiment settings

To ensure fair comparison, GCL-ALG reports the average results over five runs with different random seeds. The model is implemented using the PyTorch deep learning framework and trained on NVIDIA RTX 3090 (24 GB) and NVIDIA A40 (48 GB) GPUs.

In order to evaluate the proposed method, the following three evaluation criteria are introduced, where for all metrics higher values indicate better performance:

(1) Accuracy (ACC) refers to the proportion of correctly classified samples to the total number of samples, which can effectively evaluate datasets with balanced class distributions. We adopt ACC to assess the graph classification performance in both unsupervised and semi-supervised learning tasks.

(2) F1-score is the harmonic mean of precision and recall, which is more suitable for datasets with imbalanced class distributions. Introducing F1-score into unsupervised learning and semi-supervised learning tasks can more comprehensively evaluate the model's performance on different classes.

(3) Receiver operating characteristic area under the curve (ROC-AUC) score measures the ability of a model to distinguish between positive and negative classes at all classification thresholds. Unlike metrics such as accuracy, it is relatively robust to class imbalance, making it particularly suitable for evaluating model performance on imbalanced or distribution-shifted target domains in transfer learning scenarios.

All experiments in GCL-ALG are conducted according to the settings in Table 3. We reproduce the experimental results of automated data augmentation methods (*You et al., 2021*; *Suresh et al., 2021*; *Yin et al., 2022*), the classical contrastive frameworks (*You et al., 2020*; *Sun et al., 2019*), and the Pretrain-GNN (*Hu et al., 2019*) method, where AutoGCL (*Yin et al., 2022*) reports on joint training strategies in a semi-supervised task. For

 

**Table 3 Hyper parameters detailed settings.**

| Task | Hyperparameter | Setting |
|---|---|---|
| Unsupervised | $\tau$ | 0.5 |
| | Optimizer | Adam |
| | lr | 0.001 |
| | Epoch | 100 |
| | Batch size | 128 |
| | Hidden layers | 3/5-layers GIN |
| | Embedding dimension | 128 |
| Transfer | Pretrain lr | 0.001 |
| | Pretrain batchsize | 256 |
| | Pretrain epoch | 100 |
| | Pretrain hidden layers | 5-layer GIN |
| | Pretrain embedding Dimension | 300 |
| | Finetune lr | 0.001 |
| | Finetune batchsize | 32 |
| | Finetune epoch | 50 |
| | Finetune hidden layers | 5-layer GIN |
| | Finetune embedding Dimension | 300 |
| | Finetune dropout | 0.5 |
| Semi-supervised | lr | 0.001 |
| | Epoch | 100 |
| | Batch size | 128 |
| | Hidden layers | 3-layers ResGCN |
| | Embedding dimension | 128 |

AD-GCL (*Suresh et al., 2021*), we use the model with dynamically adjusted regularization parameters, *i.e.*, AD-GCL-OPT. And for GraphCL (*You et al., 2020*), we adopt the default augmentation strategy, *random4*.

## Experimental results and analysis

### Unsupervised learning

In this subsection, we provide an in-depth analysis of the performance of the view generator in graph classification tasks under an unsupervised learning framework. Table 4 presents a detailed comparison of our method against other mainstream methods on seven datasets in terms of classification accuracy (ACC) and F1-score. The following is our specific analysis based on the experimental results:

- Compared with classical graph contrastive learning methods (*e.g.*, InfoGraph (*Sun et al., 2019*), GraphCL (*You et al., 2020*)), GCL-ALG demonstrates significant performance advantages on most datasets. This improvement stems from the fact that traditional methods rely on fixed augmentation parameters (such as edge deletion rate and attribute masking probability), which are insufficiently adaptable to the structural characteristics of different graph data. In contrast, our learnable augmenter adjusts augmentation

**Table 4 The mean ± std experimental results for the unsupervised graph classification task based on the TuDataset benchmark datasets are shown below.** Bold numbers and underlined numbers indicate the best and second-best results, respectively. A.R. stands for average rank.

| Dataset | Metric | InfoGraph | GraphCL | JOAOv2 | AD-GCL | AutoGCL | SimGRACE | RGCL | GCL-ALG |
|---------|--------|-----------|---------|--------|--------|---------|----------|------|---------|
| MUTAG | ACC | 89.10 ± 1.48 | 88.99 ± 0.92 | 88.14 ± 0.88 | 88.31 ± 1.68 | 89.03 ± 0.73 | 87.26 ± 1.25 | 87.21 ± 0.68 | **90.37 ± 0.27** |
| | F1 | 88.09 ± 0.64 | 87.75 ± 0.74 | 87.46 ± 0.81 | 85.67 ± 2.02 | 88.37 ± 1.23 | 85.77 ± 1.96 | 86.29 ± 0.78 | **89.44 ± 0.37** |
| PROTEINS | ACC | 75.53 ± 0.74 | 75.04 ± 0.35 | 74.79 ± 0.50 | 73.31 ± 0.75 | 75.46 ± 0.54 | 74.41 ± 0.51 | 74.67 ± 0.33 | **76.33 ± 0.35** |
| | F1 | 74.55 ± 0.52 | 74.07 ± 0.50 | 73.83 ± 0.42 | 71.85 ± 0.49 | 75.08 ± 0.48 | 73.71 ± 0.72 | 73.65 ± 0.32 | **75.57 ± 0.54** |
| DD | ACC | 77.28 ± 1.83 | 78.07 ± 0.34 | **78.89 ± 0.32** | 74.68 ± 0.58 | 77.63 ± 0.85 | 76.72 ± 0.66 | 77.67 ± 0.31 | 77.73 ± 0.72 |
| | F1 | 77.08 ± 0.57 | 77.63 ± 0.70 | 77.77 ± 0.64 | 74.00 ± 0.69 | 76.69 ± 0.55 | 75.93 ± 1.10 | 76.38 ± 0.94 | **79.87 ± 0.78** |
| NCI1 | ACC | 79.24 ± 1.00 | 78.84 ± 0.31 | 79.32 ± 0.24 | 69.43 ± 0.68 | **81.52 ± 0.48** | 77.09 ± 2.17 | 77.38 ± 0.71 | 80.74 ± 0.29 |
| | F1 | 79.29 ± 0.39 | 78.66 ± 0.52 | 73.83 ± 0.42 | 68.87 ± 0.35 | **81.24 ± 0.52** | 77.58 ± 0.73 | 77.04 ± 0.21 | 80.69 ± 0.23 |
| COLLAB | ACC | 71.04 ± 3.11 | 70.98 ± 0.29 | 72.75 ± 0.78 | **73.23 ± 0.98** | 68.65 ± 2.90 | 69.62 ± 3.12 | 70.92 ± 0.88 | 72.16 ± 0.54 |
| | F1 | 66.06 ± 0.94 | 66.58 ± 1.02 | 68.33 ± 1.03 | **69.73 ± 0.85** | 62.21 ± 8.08 | 64.20 ± 5.65 | 67.64 ± 0.81 | 69.24 ± 1.07 |
| IMDB-B | ACC | 72.32 ± 1.85 | 71.32 ± 0.65 | 71.70 ± 0.57 | 71.14 ± 0.72 | 73.26 ± 2.27 | 69.90 ± 3.54 | 71.04 ± 0.49 | **74.38 ± 0.63** |
| | F1 | 72.06 ± 0.43 | 71.22 ± 0.54 | 71.38 ± 0.51 | 70.57 ± 0.76 | 71.85 ± 2.18 | 70.20 ± 0.79 | 70.11 ± 0.68 | **74.23 ± 0.56** |
| REDDIT-M-5K | ACC | 55.24 ± 3.11 | **55.72 ± 0.21** | 55.33 ± 0.20 | 54.62 ± 0.70 | 52.62 ± 4.57 | 54.54 ± 4.11 | 55.25 ± 0.73 | 55.35 ± 0.44 |
| | F1 | **55.29 ± 0.29** | 55.04 ± 0.46 | 54.86 ± 0.26 | 53.01 ± 0.61 | 51.44 ± 4.42 | 54.75 ± 0.78 | 55.03 ± 0.64 | 55.13 ± 1.29 |
| A.R. | ACC | 3.7 | 3.7 | 3.4 | 6.0 | 4.2 | 7.1 | 5.8 | 1.8 |
| | F1 | 3.1 | 4.0 | 4.4 | 6.5 | 4.1 | 6.4 | 5.8 | 1.4 |

strategies through parameter optimization, enabling the model to better capture the intrinsic properties and distributional features of graph data.

- Compared with four advanced automated data augmentation methods (JOAO (*You et al., 2021*), AD-GCL (*Suresh et al., 2021*), AutoGCL (*Yin et al., 2022*), RGCL (*Li et al., 2022*)), GCL-ALG achieves the best overall performance, with average rankings of 1.8 in ACC and 1.4 in F1-score. Existing automated methods typically focus on single-dimensional information, lacking the integration of multi-level features. Our method explicitly identifies critical nodes and edges through a centrality-driven mechanism, preserving discriminative substructures while applying appropriate perturbations, thereby effectively balancing structural integrity and representational effectiveness in augmented views. Moreover, the dual-level feature extraction framework captures both global and local structural and feature information, ensuring that augmentation is performed on reliable features and reducing the risk of information loss and semantic drift.

- Compared with the augmentation-free method (SimGRACE (*Xia et al., 2022*)), GCL-ALG consistently outperforms across all datasets, with average improvements of 2.5% in ACC and 3.1% in F1-score. This result indicates that although SimGRACE (*Xia et al., 2022*) avoids potential semantic distortion by eliminating augmentation, its reliance solely on encoder perturbation limits its ability to capture essential structural features. In contrast, GCL-ALG adopts an autoencoder-driven trend sampling strategy to generate diverse augmented views, which is more effective in enhancing the expressiveness of graph representations. Additionally, GCL-ALG exhibits significantly lower standard deviation than SimGRACE (*Xia et al., 2022*), suggesting higher

**Table 5 The results of transfer learning experiments for predicting biochemical molecular properties (mean ROC-AUC ± std).** Bold numbers represent the best performance and underlined numbers denote the second-best performance.

| Method | BBBP | Tox21 | Toxcast | SIDER | ClinTox | MUV | HIV | BACE | A.R. |
|---|---|---|---|---|---|---|---|---|---|
| *No Pretrain* | 65.8 ± 4.5 | 74.0 ± 0.8 | 63.4 ± 0.6 | 57.3 ± 1.6 | 58.0 ± 4.4 | 71.8 ± 2.5 | 75.3 ± 1.9 | 70.1 ± 5.4 | – |
| Infomax | 71.08 ± 0.57 | 75.78 ± 0.71 | 63.31 ± 0.38 | 59.96 ± 0.55 | 73.56 ± 1.96 | **79.95 ± 1.55** | 78.92 ± 0.44 | 78.47 ± 1.71 | 6.5 |
| EdgePred | 73.11 ± 0.46 | 76.37 ± 0.34 | 64.32 ± 0.36 | 61.90 ± 0.64 | 71.15 ± 1.36 | 77.49 ± 0.62 | 78.28 ± 0.64 | **82.73 ± 1.21** | 4.3 |
| AttrMasking | 72.17 ± 0.84 | **77.22 ± 0.49** | 64.69 ± 0.65 | 61.99 ± 0.64 | 80.82 ± 5.39 | 78.59 ± 1.41 | 78.59 ± 0.47 | 80.19 ± 0.74 | 3.6 |
| ContextPred | 73.21 ± 0.55 | 76.10 ± 0.49 | 64.26 ± 0.33 | 61.45 ± 0.60 | 71.32 ± 1.08 | 79.46 ± 0.78 | 78.71 ± 0.41 | 81.97 ± 1.12 | 4.2 |
| GraphCL | 70.96 ± 0.88 | 74.65 ± 0.67 | 62.24 ± 0.52 | 61.24 ± 0.71 | 63.68 ± 2.05 | 75.10 ± 1.42 | 76.91 ± 1.49 | 81.00 ± 1.20 | 9.0 |
| JOAOv2 | 73.17 ± 0.76 | 75.72 ± 0.48 | 64.17 ± 0.53 | 62.68 ± 0.38 | 79.56 ± 2.28 | 76.78 ± 1.07 | 78.01 ± 1.01 | 79.12 ± 1.20 | 5.6 |
| AD-GCL | 68.49 ± 1.21 | 76.12 ± 0.36 | 62.52 ± 0.12 | 59.98 ± 0.41 | 73.71 ± 1.67 | 75.64 ± 0.91 | 77.36 ± 0.83 | 81.24 ± 1.06 | 7.6 |
| AutoGCL | 70.84 ± 0.92 | 76.14 ± 0.56 | 63.76 ± 0.34 | **63.70 ± 0.76** | 77.75 ± 2.90 | 76.77 ± 1.17 | 78.05 ± 0.67 | 79.69 ± 0.45 | 5.7 |
| SimGRACE | 70.51 ± 0.94 | 74.78 ± 0.76 | 62.41 ± 0.63 | 60.91 ± 0.51 | 60.95 ± 2.02 | 75.45 ± 2.07 | 76.97 ± 0.71 | 81.21 ± 1.59 | 9.1 |
| RGCL | 71.37 ± 0.62 | 74.98 ± 0.18 | 63.21 ± 0.26 | 60.95 ± 0.34 | 86.47 ± 0.65 | 73.51 ± 0.59 | 76.08 ± 0.72 | 75.99 ± 0.24 | 8.2 |
| GCL-ALG | **74.48 ± 0.78** | 76.54 ± 0.31 | **65.03 ± 0.25** | 62.71 ± 0.97 | **87.33 ± 0.91** | 77.52 ± 0.90 | **79.09 ± 0.62** | 82.00 ± 1.14 | 1.7 |

robustness and confirming the effectiveness of well-designed learnable augmentations over augmentation-free approaches.

- Dataset-based analysis. GCL-ALG consistently ranks among the top three across all datasets, showing excellent performance and strong generalizability across different types of datasets. In social networks, where high connectivity and complex structures may cause topological distortion under augmentation, and in biochemical molecules, where structural links carry chemical semantics, GCL-ALG's centrality-guided strategy effectively identifies and preserves key substructures, leading to improved performance. Furthermore, GCL-ALG maintains stable and superior results across datasets ranging from the small-scale MUTAG (188 graphs) to the large-scale REDDIT-M-5K (4,999 graphs), further validating its strong robustness to varying graph scales.

### Transfer learning

To further validate the broad applicability of the proposed algorithm, we conduct transfer learning experiments to evaluate its performance on biochemical molecular property prediction tasks. Following the setup in GraphCL (*You et al., 2020*), we first pretrain the graph view generators on the large-scale biomolecule dataset CHEMBL (*Mayr et al., 2018*; *Gaulton et al., 2011*), and then fine-tune it on several downstream chemical molecule datasets of varying scales. As shown in Table 5, GCL-ALG consistently demonstrates superior performance across all target datasets, achieving an average ranking of 1.7, outperforming all baseline methods. Specifically, GCL-ALG achieves the best results on the blood-brain barrier penetration (BBBP) dataset, US EPA Toxicity Forecaster (ToxCast), ClinTox, and HIV, with average performance improvements of 2.97%, 1.54%, 13.43%, and 1.3%, respectively. This indicates that the view generators in GCL-ALG are capable of successfully learning diverse graph features and intrinsic structural patterns during the pretraining phase, and the learned representations demonstrate strong generalization ability during the fine-tuning stage.

**Table 6** The semi-supervised graph classification performance on the TuDataset benchmark datasets with 10% labeled data. Bold numbers indicate the best performance and underlined numbers indicate the second-best performance.

| Dataset | Metric | Full data | 10% Data | GCA | GraphCL | JOAOv2 | AD-GCL | AutoGCL | SimGRACE | GPA | GCL-ALG |
|---|---|---|---|---|---|---|---|---|---|---|---|
| PROTEINS | ACC | 78.68 ± 0.50 | 73.19 ± 1.63 | 73.85 ± 5.56 | 72.67 ± 1.53 | 75.14 ± 1.87 | 73.96 ± 0.47 | 75.02 ± 0.32 | 71.09 ± 0.91 | 73.18 ± 0.17 | **75.78 ± 0.62** |
|  | F1 | 77.40 ± 0.45 | 72.13 ± 1.45 | 72.23 ± 1.35 | 73.07 ± 0.68 | 71.93 ± 1.05 | 71.04 ± 0.45 | 74.48 ± 0.39 | 70.45 ± 0.96 | 71.89 ± 0.72 | **75.26 ± 0.29** |
| DD | ACC | 81.59 ± 0.26 | 76.78 ± 1.14 | 76.74 ± 4.09 | 75.48 ± 1.02 | 74.88 ± 1.10 | 77.91 ± 0.73 | 77.64 ± 0.52 | 74.50 ± 0.89 | 76.24 ± 0.31 | **78.82 ± 4.71** |
|  | F1 | 80.58 ± 1.12 | 75.89 ± 1.12 | 74.94 ± 0.55 | 74.35 ± 1.25 | 74.19 ± 0.99 | 74.82 ± 0.37 | 77.32 ± 0.53 | 73.70 ± 0.81 | 75.19 ± 0.15 | **78.54 ± 1.02** |
| NCI1 | ACC | 84.90 ± 0.27 | **76.00 ± 0.45** | 68.73 ± 2.36 | 74.12 ± 0.55 | 73.77 ± 0.51 | 75.18 ± 0.31 | 70.01 ± 0.37 | 73.48 ± 0.56 | 74.14 ± 0.38 | 68.92 ± 0.46 |
|  | F1 | 84.85 ± 0.30 | **75.89 ± 0.46** | 67.65 ± 2.23 | 74.46 ± 0.42 | 73.62 ± 0.47 | 73.66 ± 0.63 | 69.79 ± 0.57 | 73.15 ± 0.52 | 72.69 ± 0.02 | 69.41 ± 0.32 |
| COLLAB | ACC | 83.49 ± 0.19 | 75.01 ± 0.63 | 74.32 ± 2.30 | 73.83 ± 0.36 | 75.14 ± 0.33 | 75.82 ± 0.26 | **77.45 ± 0.32** | 74.02 ± 0.43 | 70.17 ± 0.19 | 76.73 ± 0.31 |
|  | F1 | 81.05 ± 0.36 | 71.57 ± 0.85 | 70.17 ± 2.09 | 70.57 ± 0.34 | 71.97 ± 0.39 | 70.99 ± 0.41 | 73.54 ± 0.46 | 70.45 ± 0.68 | 68.37 ± 0.79 | **73.87 ± 0.84** |
| REDDIT-M-5K | ACC | 56.27 ± 2.77 | 34.63 ± 0.96 | 32.95 ± 10.89 | 52.33 ± 0.54 | 50.90 ± 0.63 | 53.49 ± 0.28 | 37.11 ± 1.04 | 50.81 ± 0.46 | **53.76 ± 0.24** | 51.14 ± 0.75 |
|  | F1 | 53.26 ± 2.53 | 33.77 ± 0.48 | 28.91 ± 9.59 | **52.22 ± 0.32** | 51.76 ± 0.65 | 50.35 ± 0.33 | 29.18 ± 1.22 | 48.69 ± 0.44 | 52.08 ± 0.59 | 48.96 ± 0.98 |
| A.R. | ACC | – | 4.8 | 6.8 | 6.0 | 4.8 | 2.6 | 4.2 | 7.4 | 5.3 | 3.2 |
|  | F1 | – | 4.4 | 7.0 | 3.8 | 4.8 | 5.2 | 4.0 | 7.2 | 5.4 | 3.2 |

### Semi-supervised learning

Finally, we evaluate the semi-supervised classification performance of baseline approaches and GCL-ALG using 10-fold cross-validation. According to the experimental results shown in Table 6, we have the following analyses:

- **Robustness performance**. GCL-ALG achieves training performance close to that with full supervision even when using only 10% labeled data. Notably, on the PROTEINS and DD datasets, the performance gap remains within 3%, highlighting the model's strong feature extraction capabilities and robustness under data scarcity.

- **Advantages of the view generator**. According to experimental results, GCL-ALG achieves an outstanding average rank of 2.6 in terms of ACC and F1-score. This advantage stems from its context-aware adaptive augmentation strategy, which aligns well with the intrinsic structure of the graphs. Compared to AutoGCL (*Yin et al., 2022*), GCL-ALG shows performance improvements ranging from 0.33% to 19.78%. Against GPA (*Zhang et al., 2024*), it achieves average gains of 0.78% in ACC and 1.16% in F1-score, validating the effectiveness of the proposed adaptive strategy.

- **Limitation analysis**. Despite its overall strong performance, GCL-ALG exhibits certain limitations in specific scenarios. On the small and sparse molecular dataset NCI1 under low label rates, edge-level augmentations may induce substantial changes in graph connectivity, leading to information loss and compromised classification performance. Furthermore, GCL-ALG demonstrates only moderate effectiveness on the REDDIT-M-5K dataset, which can be attributed to a dual sparsity problem under limited supervision—characterized by sparsity in both class distribution and label availability. This condition hinders the model's ability to learn discriminative features for minority classes, while centrality-based augmentation may be skewed toward dominant class structures, resulting in imbalanced representation learning.

- **Performance across dataset types**. Based on the experimental results from unsupervised, semi-supervised, and transfer learning, GCL-ALG demonstrates superior performance on biological and chemical molecular datasets. These graphs typically

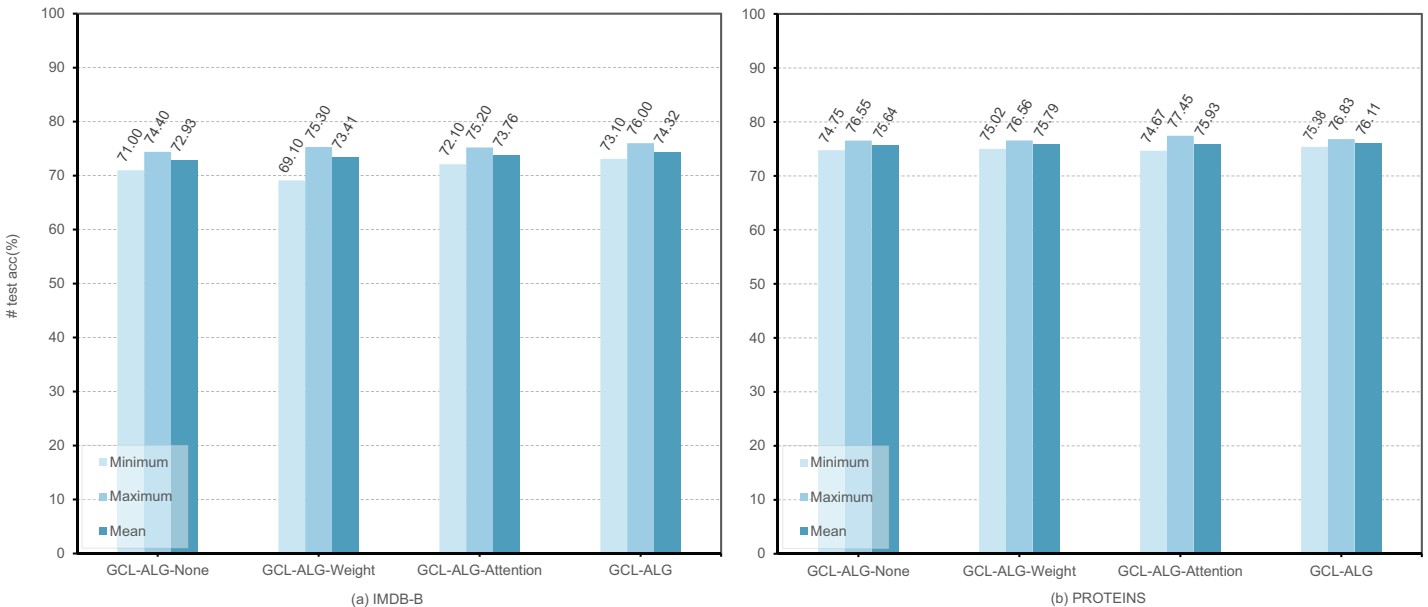

**Figure 3 The performance of different mechanisms in $\mathcal{F}_A$, $\mathcal{F}_E$ encoders.** We execute experiments under four distinct scenarios: GCL-ALG-None, GCL-ALG-Weight, GCL-ALG-Attention, and GCL-ALG to evaluate the performance impact within the context of an unsupervised learning graph classification task.                                                     

exhibit well-defined functional motifs and stable topological structures, which align well with the centrality-guided adaptive augmentation strategy employed by GCL-ALG. In contrast, social network graphs exhibit more complex and hierarchical structures, where a single centrality measure is insufficient to capture the multifaceted information, thereby limiting performance improvements relatively. Future work may explore the integration of multiple centrality metrics or community structural features to further enhance the model's ability to represent and learn from social network data.

## Detailed analysis of the encoder mechanism

To systematically evaluate the effectiveness of the dual-layer feature extraction framework and the synergy between its constituent modules, we conduct ablation experiments focusing on the independent contributions and interactive effects of structural modeling strategies based on self-attention mechanisms and edge load information. The experiments employ an SVM classifier, with training repeated 10 times under different random seeds. In each run, the unsupervised graph representation learning process is iterated for 100 epochs to ensure the stability and reliability of the results. Figure 3 reports the maximum, minimum, and average performance of GCL-ALG and its structural variants on two representative graph classification datasets, IMDB-B and PROTEINS. The variants include GCL-ALG-Attention, which contains only the self-attention-based GNN encoder; GCL-ALG-Weight, which includes only the edge-based GNN encoder; and GCL-ALG-None, which excludes both components. The results yield the following insights:

- **Synergistic enhancement effect**. From Fig. 3, it can be observed that for methods GCL-ALG-None, GCL-ALG-Weight, GCL-ALG-Attention, and GCL-ALG, their

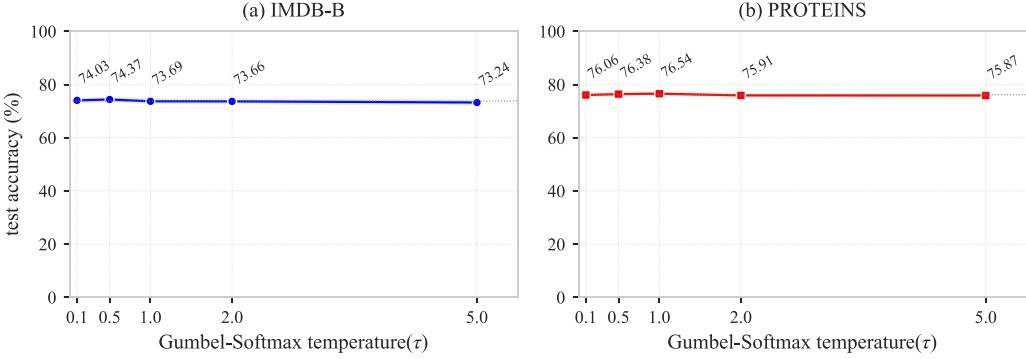

**Figure 4 Sensitivity analysis of the Gumbel-Softmax hyperparameter $\tau$.**

classification performances in both datasets show a progressive improvement. The GCL-ALG model, which jointly integrates both components, consistently outperforms all other variants in terms of maximum, minimum, and average performance. Specifically, GCL-ALG achieves average improvements of 1.39% on IMDB-B and 0.47% on PROTEINS compared to the GCL-ALG-None baseline. This not only validates the individual effectiveness of each component, but also highlights the positive complementarity between them.

- **Differential module contributions**. The experimental results reveal that GCL-ALG-Attention yields average gains of 0.83% and 0.29% on IMDB-B and PROTEINS, respectively, while GCL-ALG-Weight achieves only 0.48% and 0.15%. This discrepancy indicates that, in the context of unsupervised graph representation learning, the dynamically adaptive attention mechanism offers superior discriminative capacity compared to static structural measures such as edge load, enabling more effective identification of local structural features relevant to the task.

## Hyperparameter analysis

In this section, we analyze the sensitivity of the Gumbel-Softmax temperature parameter $\tau$ in the GCL-ALG algorithm. $\tau$ controls the sharpness of the edge sampling probability distribution in the learnable view generator, directly affecting the balance between the diversity and the information completeness of the generated views. To explore the impact of $\tau$ on different types of graph data, we conduct experiments with $\tau \in 0.1, 0.5, 1.0, 2.0, 5.0$, using two representative datasets: IMDB-B and PROTEINS. As shown in Fig. 4, the IMDB-B dataset achieves the best performance of 74.37% when $\tau = 0.5$, after which the performance gradually decreases as $\tau$ increases. The PROTEINS dataset reaches its peak performance of 76.54% at $\tau = 1.0$ and maintains relatively stable performance within the range $\tau \in [0.1, 1.0]$, but shows a significant performance drop when $\tau \geq 2.0$. The analysis indicates that IMDB-B performs best at a moderate $\tau$ value, suggesting that an appropriate degree of randomness helps alleviate the issue of local structure dependency. As a biologically meaningful graph dataset rich in features, PROTEINS maintains stable performance within the range $\tau \in [0.1, 1.0]$, but exhibits a decline when $\tau \geq 2.0$, indicating

that excessive randomness disrupts its inherent feature-structure correlations. It is worth noting that both datasets show a performance drop when $\tau \geq 2.0$, confirming that an overly high temperature introduces excessive noise, which weakens the effective gradient signal and affects the learning of the view generator. Overall, except for the performance degradation at higher temperature values, the model exhibits limited sensitivity to $\tau$ and good robustness, with the regularization and fully connected structures also contributing to mitigating performance fluctuations. Based on the above experimental results, we recommend setting the temperature parameter $\tau$ within the range $[0.5, 1.0]$.

## CONCLUSION

In this article, we propose a novel data augmentation approach to address the challenges posed by the tremendous heterogeneity of graph structure data and the constraints of existing data augmentation strategies. The approach is rooted in learning probability distributions from adaptive graph structures, creating dynamic, learning, adaptive view generators that efficiently extract graph features relevant to downstream tasks. To achieve this goal, GCL-ALG applies the attention mechanism and edge load information to two distinct GNN encoders and optimizes the adaptive sampling strategy under the guidance of the network centrality metric. Extensive analysis and experiments demonstrate the superiority of our proposed GCL-ALG approach over other automated data augmentation methods on multiple tasks. Furthermore, we investigate the individual contributions as well as the combined effects of the attention mechanism and edge weights, and the results reveal a synergistic relationship between them.

In future research, attributes will be integrated into learnable feature mining, aiming to the ability of the model to learn relationships between attributes and underlying graph structures, thereby achieving more effective and comprehensive graph representation learning. Meanwhile, optimizing samples by incorporating neighborhood information is also worthy of in-depth exploration.

### Funding

This work was supported by the National Natural Science Foundation of China under Grant No. 62006009, Beijing Natural Science Foundation No. L233005 and Beijing Natural Science Foundation No. 4244072. The funders had no role in study design, data collection and analysis, decision to publish, or preparation of the manuscript.

### Grant Disclosures

The following grant information was disclosed by the authors:
National Natural Science Foundation of China: 62006009.
Beijing Natural Science Foundation: L233005 and 4244072.

### Competing Interests

The authors declare that they have no competing interests.

## Author Contributions

- Yafang Li conceived and designed the experiments, analyzed the data, authored or reviewed drafts of the article, and approved the final draft.
- Jie Kang conceived and designed the experiments, performed the experiments, analyzed the data, performed the computation work, prepared figures and/or tables, authored or reviewed drafts of the article, and approved the final draft.
- Zhihua Chu performed the computation work, authored or reviewed drafts of the article, and approved the final draft.
- Baokai Zu conceived and designed the experiments, authored or reviewed drafts of the article, and approved the final draft.

## Data Availability

The code is available at GitHub:

- https://github.com/lincyli/GCL-ALG

- lincyli. (2025). lincyli/GCL-ALG: GCL-Algorithm v2.0.0 (v2.0.0). Zenodo. https://doi.org/10.5281/zenodo.16731646

The TDdatasets are available at: https://chrsmrrs.github.io/datasets/docs/datasets.

The MoleculeNet datatset is available at: https://moleculenet.org/datasets-1.

The Chembel datatset is available at: https://www.ebi.ac.uk/chembl.

## Supplemental Information

Supplemental information for this article can be found online at http://dx.doi.org/10.7717/peerj-cs.3101#supplemental-information.

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
