# Peer review of "GCL-ALG: graph contrastive learning with adaptive learnable view generators"

_PeerJ Computer Science, doi:10.7717/peerj-cs.3101_

## Round 0.1 · original submission · Major Revisions

Please address the reviewer comments in your next revision.

**Language Note:** The review process has identified that the English language must be improved. PeerJ can provide language editing services - please contact us at [email protected] for pricing (be sure to provide your manuscript number and title). Alternatively, you should make your own arrangements to improve the language quality and provide details in your response letter. – PeerJ Staff

Reviewer 1 ·

Basic reporting

GCL-ALG is a novel graph contrastive learning framework that employs adaptive learnable view generators. It uses a dual-level feature extraction approach with GNNs, attention mechanisms, and edge probabilities to generate reliable graph representations. The framework then leverages network science to selectively modify augmentation probabilities at the node and edge levels, enabling the training of dynamic augmentation instances.

Experimental design

The experimental section of this work extensively evaluates the proposed GCL-ALG framework on 10 benchmark datasets across unsupervised learning, semi-supervised learning, and transfer learning tasks. The results demonstrate the superior performance of GCL-ALG compared to state-of-the-art baseline methods, with the framework exhibiting outstanding average relative improvement.

Validity of the findings

The key experimental findings of this work demonstrate the superior performance of the proposed GCL-ALG framework. GCL-ALG outperformed state-of-the-art methods across a range of benchmark datasets and tasks, including unsupervised learning, semi-supervised learning, and transfer learning. The authors' ablation studies further revealed the significant contribution of the dual-level feature extraction approach to the algorithm's strong performance.

Additional comments

The authors should consider expanding on the technical details of the dual-level feature extraction framework proposed in GCL-ALG. While the general concept is described, more specifics on the implementation and rationale behind the two levels would be beneficial.

For example, the authors could delve deeper into the motivations for combining self-attention networks with GNN encoders in the first level, and how the integration of edge load information in the second level complements the representation learning.

The related work section of the paper could be strengthened by including a more comprehensive review of recent advancements in graph contrastive learning frameworks. Specifically, the authors may want to consider discussing works such as "Candidate-aware Graph Contrastive Learning for Recommendation" (SIGIR 2023), and "Adaptive Graph Contrastive Learning Recommendation" (KDD 2023).

Cite this review as

Reviewer 2 ·

Basic reporting

In general, the work presented in this paper is very good and have substantial evidence of an impactful contribution.
The work presented are very clear and the English usage is very good.
Literature used are sufficient and the article are properly done and very structured. Figures and tables presented looks very good.

Experimental design

The experimental design is very good, well planned and well described.

However, I would want to know why only 3 domains (social network, biochemical and chemical) are chosen. Please describe the justification for it.

Validity of the findings

One of the glaring ambiguities within this work is the use of datasets which are totally different in terms of
of the contribution. As far as I am aware, your work is trying to put forward the generality of data augmentation based on the learning concept regardless of the data domain. In order to describe that your approach is in a way good (or not good) for any type of data (domain), I think there must be an analysis of the effect of your approach from the perspective of the data itself. For example, you had 3 types of datasets namely social network, biochemical network and chemical modules for your experiment. I think it is better to explain all the different impact and effect (I assume) on these datasets as each of these datasets have different nodes, edges and classes. For example, you found out that your proposed work did give favorable result on the NC11 dataset. Based on the information given, NC11 dataset has among the lowest graphs, average nodes, and classes and medium number of averages nodes. If this is relevant, please explain how this has impact towards the study.

In page 15, you did mention about your joint based method performance surpasses (on average) the non-joint method by only 1.39% and 0.47% respectively. How is this can be claimed as notable advantage especially for the 0.47% increase?

Additional comments

There are a couple of mistakes by the term tudataset when instead it should be TUDataset.

Cite this review as

Reviewer 3 ·

Basic reporting

The research addresses a well-defined problem in graph contrastive learning by introducing an adaptive learnable augmentation technique. The proposed approach effectively tackles the limitations of predefined augmentation strategies.

Minor:
Figures are clear and well-labeled, but Figure 1 could benefit from additional explanation in the text to enhance interpretability.
There are a few minor grammatical and typographical errors that should be corrected for improved readability. For instance, in the introduction, "achevements" should be corrected to "achievements."

Major:
The article presents a well-structured study with clear research motivation and a coherent theoretical foundation. However, the related work section lacks recent studies from 2023, which are crucial to benchmarking the proposed approach. Several high-impact works in Graph Contrastive Learning (GCL) and Graph Neural Networks (GNNs) are missing, particularly those with higher accuracy. Thus, the paper does not compare its proposed GCL-ALG method against more recent works from 2023, many of which have demonstrated superior accuracy in graph contrastive learning. Notable missing comparisons include:
1. Mixup for Node and Graph Classification: This method explores interpolation-based augmentation, which has been shown to be effective in improving classification tasks.
2. SimGRACE: A Simple Framework for Graph Contrastive Learning Without Data Augmentation—This work challenges the necessity of traditional augmentation techniques and achieves competitive performance without data augmentation, directly questioning the fundamental assumption of GCL-ALG.
3. TGNN: A Joint Semi-Supervised Framework for Graph-Level Classification—A more recent semi-supervised approach that integrates both supervised and unsupervised contrastive learning strategies.
4. Graph Contrastive Multi-view Learning: A Pre-Training Framework for Graph Classification—This work explores multi-view learning strategies to enhance contrastive learning, potentially achieving better accuracy than GCL-ALG.

I suggest authors:
1. Expand the related work section to discuss these more recent methods and explain how GCL-ALG differentiates itself.
2. Clearly highlight the novelty of the proposed approach in comparison to state-of-the-art methods.
3. Conduct additional experiments to compare GCL-ALG against these state-of-the-art approaches. If it is computationally infeasible to rerun all experiments, at least provide a discussion of their originally reported accuracy versus that of GCL-ALG.
4. If GCL-ALG underperforms these newer methods, justify why it is still relevant (in terms of efficiency, interpretability, and adaptability).
5. Update the reference list to include key studies to demonstrate awareness of recent advancements.

Experimental design

While the paper presents a structured methodology, there are several areas where it falls short in meeting publication standards. Below are key concerns and suggested improvements:

1. The authors should discuss Graph Attention Networks (GAT) and Graph Isomorphism Networks (GIN) and what they mean by "The view generator is built upon two GNN encoders with GIN as the backbone, namely the attention-based GNN encoder." GAT is built on the principle that the approach automatically learns the importance of each neighbor based on the attention mechanism; therefore, the fundamental question is: What is new in the attention-based GNN encoder? Why GIN when we already have GAT? One may be of the view that using GAT as an encoder may be computationally cheaper and perform more accurately than integrating a self-attention network module after the GIN layer. Hence, let's have an apparent reason using GIN.

2. The authors may consider providing more details on the implementation process and specifying the evaluation metrics used rather than restating the formulas for accuracy (ACC) and F1-score. While including these formulas may be helpful for beginners, they do not significantly enhance the overall impact of the report.

3. The authors also stated, "The ROC curve was plotted with True Positive Rate (TPR = TP/(TP+FN)) as the vertical axis and False Positive Rate (FPR = FP/(FP+TN)) as the horizontal axis, and the area under the curve (AUC) was calculated by numerical integration." However, no such plot appears to be included in the paper. It would be helpful to clarify this discrepancy and ensure that the referenced figure is either properly included or appropriately described within the text.

4. To further enrich the discussion on data augmentation in GNNs, the authors may consider citing the paper ‘Towards Data Augmentation in Graph Neural Networks: An Overview and Evaluation’. This paper provides a comprehensive overview of data augmentation techniques in GNNs, categorizing them into node, edge, and structure-based modifications. Additionally, it highlights the impact of these techniques on model performance and generalization, which could complement the findings presented in this study.

Validity of the findings

No comment

Cite this review as

---

## Round 0.2 · Minor Revisions

Reviewer 1 still has some comments that must be addressed.

Reviewer 1 ·

Basic reporting

This paper introduces GCL-ALG, a novel approach for graph contrastive learning that employs adaptive learnable view generators. The method addresses limitations of existing graph augmentation strategies by using a dual-level feature extraction framework that combines graph neural networks with attention mechanisms and edge probability distributions.

Experimental design

The authors evaluated GCL-ALG across three learning paradigms: unsupervised, semi-supervised, and transfer learning. They used 16 benchmark datasets spanning social networks, biochemical molecules, and chemical compounds. For unsupervised learning, they employed SVM classifiers on node embeddings.

Validity of the findings

The experimental results appear statistically sound with repeated runs and standard deviation reporting. GCL-ALG consistently outperforms baseline methods across multiple datasets, with comprehensive ablation studies confirming the contributions of its key components.

Additional comments

The paper currently relies on degree centrality as the primary guidance signal for node/edge sampling. The authors should expand this discussion to explore how alternative centrality measures (betweenness, closeness, eigenvector centrality) might better capture structural importance in different graph types.

The paper mentions the temperature parameter τ in the Gumbel-Softmax sampling but doesn't adequately explore its impact on augmentation diversity and model performance. A sensitivity analysis showing how different temperature values affect the trade-off between exploration (diverse views) and exploitation (semantic preservation) would provide valuable insights.

While the paper provides time complexity analysis, it lacks empirical runtime comparisons with baseline methods and scalability testing on larger graphs.

Cite this review as

Reviewer 2 ·

Basic reporting

The paper has shown tremendous improvement in terms of the language used, and many more corrections that have been made based on the rebuttal documents.

Experimental design

The experimental design of the study is well-structured and methodologically sound. The authors have clearly defined their research objectives and formulated appropriate hypotheses. The selection of variables, control measures, and experimental conditions is suitable to address the research questions effectively. Additionally, the authors have employed robust data collection methods and appropriate sampling strategies that enhance the reliability and validity of the findings. The study's experimental procedures are well-documented, allowing for reproducibility and transparency. Overall, the design is logical, systematic, and contributes to the credibility of the results

Validity of the findings

The findings of the study are well-supported by the data, and the analysis is appropriate and thorough. The authors have taken steps to ensure the results are reliable and can be trusted. The conclusions are clearly linked to the evidence presented, making the study's claims credible and valid.

Cite this review as

Reviewer 3 ·

Basic reporting

Literature references and sufficient field background/context provided.
Professional article structure, figures, and tables. Raw data shared.
Self-contained with relevant results to hypotheses.

Experimental design

Original primary research within the Aims and Scope of the journal.
Research question well defined, relevant & meaningful.
The authors stated how research fills an identified knowledge gap.
Rigorous investigation performed to a high technical & ethical standard.
Methods are described with sufficient detail & information to replicate.

Validity of the findings

All underlying data have been provided; they are robust, statistically sound, and controlled.
Conclusions are well stated, linked to the original research question, and limited to supporting results.

Additional comments

The authors have adequately addressed my concerns regarding the paper. I recommend acceptance.

Cite this review as

---

## Round 0.3 · accepted · Accept

I believe the authors have addressed the concerns of the reviewers. Thanks for their efforts.